# Effect of the Content and Ordering of the sp^2^ Free Carbon Phase on the Charge Carrier Transport in Polymer-Derived Silicon Oxycarbides

**DOI:** 10.3390/molecules25245919

**Published:** 2020-12-14

**Authors:** Felix Rosenburg, Benjamin Balke, Norbert Nicoloso, Ralf Riedel, Emanuel Ionescu

**Affiliations:** 1Institut für Material- und Geowissenschaften, Technische Universität Darmstadt, Otto-Berndt-Straße 3, 64287 Darmstadt, Germany; f.rosenburg@gmx.de (F.R.); nn@eurad.de (N.N.); ralf.riedel@tu-darmstadt.de (R.R.); 2Institut für Anorganische Chemie und Analytische Chemie, Johannes-Gutenberg-Universität Mainz, Duesbergweg 10-14, 55128 Mainz, Germany; benjamin.balke@iwks.fraunhofer.de; 3Fraunhofer Research Institution for Materials Recycling and Resource Strategies IWKS, Rodenbacher Chaussee 4, 63457 Hanau, Germany

**Keywords:** silicon oxycarbides, charge carrier transport, free carbon, Raman spectroscopy, Hall measurements

## Abstract

The present work elaborates on the correlation between the amount and ordering of the free carbon phase in silicon oxycarbides and their charge carrier transport behavior. Thus, silicon oxycarbides possessing free carbon contents from 0 to ca. 58 vol.% (SiOC/C) were synthesized and exposed to temperatures from 1100 to 1800 °C. The prepared samples were extensively analyzed concerning the thermal evolution of the sp^2^ carbon phase by means of Raman spectroscopy. Additionally, electrical conductivity and Hall measurements were performed and correlated with the structural information obtained from the Raman spectroscopic investigation. It is shown that the percolation threshold in SiOC/C samples depends on the temperature of their thermal treatment, varying from ca. 20 vol.% in the samples prepared at 1100 °C to ca. 6 vol.% for the samples annealed at 1600 °C. Moreover, three different conduction regimes are identified in SiOC/C, depending on its sp^2^ carbon content: (i) at low carbon contents (i.e., <1 vol.%), the silicon oxycarbide glassy matrix dominates the charge carrier transport, which exhibits an activation energy of ca. 1 eV and occurs within localized states, presumably dangling bonds; (ii) near the percolation threshold, tunneling or hopping of charge carriers between spatially separated sp^2^ carbon precipitates appear to be responsible for the electrical conductivity; (iii) whereas above the percolation threshold, the charge carrier transport is only weakly activated (E_a_ = 0.03 eV) and is realized through the (continuous) carbon phase. Hall measurements on SiOC/C samples above the percolation threshold indicate p-type carriers mainly contributing to conduction. Their density is shown to vary with the sp^2^ carbon content in the range from 10^14^ to 10^19^ cm^−3^; whereas their mobility (ca. 3 cm^2^/V) seems to not depend on the sp^2^ carbon content.

## 1. Introduction

Silicon oxycarbides have been studied for several decades and attracted special attention due to the unique way in which they exhibit and combine structural features and properties of glasses with those of interface-dominated nano-heterogeneous materials [1]. Thus, silicon oxycarbide typically can be described as silicate glasses with strong carbon-for-oxygen substitution [2,3]. The incorporation of carbon in the silica glass network has been shown in various studies to improve its properties. It was shown for instance that silicon oxycarbide glasses have significantly increased Young’s moduli, hardness values, fracture toughness and devitrification resistance when compared to vitreous silica [1]. This effect has been attributed to an increased network connectivity of the silicon oxycarbides as compared to that of vitreous silica. Moreover, studies concerning the network architecture of silicon oxycarbide glasses revealed a relatively low mass fractal dimension of those materials (typically in the range of 2.3–2.5) [2], which has been reflected for instance in a very low Poisson ratio (i.e., 0.11 as compared to 0.17 in silica-based glasses) [4] and highly promising bioactivity with respect to hydroxyapatite mineralization [5,6,7]. The glassy network of silicon oxycarbides can be easily altered via incorporating additional network formers (i.e., boron) [8] or modifiers (e.g., Li, Ca, Mg, etc.) [3,5,6,9,10], thus structural properties such as network connectivity and polymerization can be finely tuned [6]. This allows for providing finely adjustable properties in silicon-oxycarbide-based materials.

In addition to the unique structural architecture of their glassy network and their remarkable property profile correlated to it, silicon oxycarbides can be designed to contain an in situ formed sp^2^-hybridized carbon phase finely dispersed within their microstructure. This segregated carbon phase has been shown to be responsible for a plethora of interesting properties of silicon oxycarbides such as reversible Li uptake/release [11,12,13,14,15], tunable thermal and electrical transport [16,17,18], piezoresistive behavior [19,20,21], near-zero creep at high temperatures [22,23,24,25], electromagnetic wave absorption [26], tailorable surface energy [27], adjustable blood clothing effect [28], etc. The amount of sp^2^-hybridized free carbon can be varied from a negligible fraction [29] up to very large extents by adjusting the macromolecular architecture of the preceramic polymers and by considering the introduction of additional carbon sources (e.g., divinyl benzene or other aromatics-containing compounds) either by blending or by chemical bonding [1,30,31]. Moreover, the synthesis temperature of free-carbon-containing silicon oxycarbides (i.e., SiOC/C) has been shown to directly correlate to the ordering degree (crystallinity) of the free carbon phase [32]. Thus, the sp^2^-hybridized carbon phase shows a highly disordered structure at temperatures of ca. 1000–1200 °C typically used for the synthesis of SiOC/C; whereas annealing of SiOC/C at higher temperatures leads to a significant ordering of the free carbon phase [1,32].

As mentioned above, the concentration and ordering of carbon in silicon oxycarbides has been shown to directly affect their properties; for instance, the electrical conductivity in silicon oxycarbides can be progressively changed from an insulator to a semiconductor and semimetal [1]. As recently reported in a case study, SiOC/C with a carbon content of ca. 30 vol.%, which was prepared via pyrolysis of a preceramic polymer at 1000 °C showed a resistivity value of *ρ* = 0.35 Ω/m, which further decreased to *ρ* = 0.14 Ω/m upon pyrolysis at higher temperatures [33]; whereas a SiOC/C material derived from DVB-added polyhydridomethylsiloxane exhibited an even lower value of 0.23 Ω/m [34] (note that graphite has a resistivity ≤10^−3^ Ω/m). These values are significantly lower than those usually reported for SiOC/C with carbon contents below 7.5 vol.%, which are typically in the range of 10^7^–10^10^ Ω/m) [35]. The resistivity in SiOC/C is thus considered to correlate to the amount of segregated carbon, which can be altered by adjusting the macromolecular architecture of the precursor, and to its degree of ordering, which can be varied by changing the temperature of thermal treatment [36,37,38]. Depending on the synthesis conditions, a broad range of different carbon microstructures can be present in SiOC/C ceramics, covering a wide range of electronic properties. At one extreme, the electronic conduction in SiOC/C approximates that of graphite/sp^2^-carbon, whereas at the other extreme, the free carbon barely contributes to the electronic transport, which is consequently determined by the insulating SiOC glassy matrix [34,35,39]. The first border case may correspond to high amounts of the sp^2^-hybridized carbon phase and/or high crystallinity thereof; whereas the latter involves small fractions of free carbon and/or high disorder [1,40].

The present work studies the correlations between the amount and crystallinity of the free carbon phase in silicon oxycarbides and their charge carrier transport behavior. Thus, detailed structural characterization of the segregated carbon phase, electrical conductivity and Hall effect measurements will be introduced and discussed within the context of rationalizing different regimes of charge carrier transport in SiOC/C.

## 2. Results and Discussion

In order to synthesize polymer-derived SiOC/C ceramics with different compositions, following commercially available preceramic polymer resins were used: Polyramic^®^ SPR-212, SPR-684 and SPR-688 (Starfire Systems Inc., Glenville, NY, USA) and BELSIL^®^ PMS-MK and SILRES^®^ 604 (Wacker Chemie AG, Munich, Germany). Further SiOC/C compositions were obtained from precursors prepared via sol–gel processing upon using different ratios of triethoxysilane TREOS, methyldiethoxysilane MDES, triethoxymethylsilane TEMS or triethoxyvinylsilane TEVS (Merck KGaA, Darmstadt, Germany). Table 1 lists the different silicon oxycarbide compositions considered in the present study and the precursors used for their synthesis.

A brief summary of the elemental composition of the SiOC/C samples heat treated at T = 1600 °C is presented in Table 1. It is shown that the total amount of carbon present in the prepared silicon oxycarbide materials can be adjusted over a wide range, i.e., from ca. 7 (sample C-0) up to 55 wt % (sample C-60). Based on the empirical formulae of the prepared materials (Table 2), which can be described as SiO_2(1 − x)_C_x_ + y C (Table 2), it can be stated that the prepared materials show comparable contents of network carbon (i.e., x values range between 0.26 for C-11 and 0.33 for C-36), with the exception of C-17 and C-60, which show an increased network carbon content, i.e., x values of 0.53 and 0.47, respectively. In the series of silicon oxycarbide formulations prepared in the present study, the volume fraction of segregated carbon, which was estimated cf.
(1)$$\phi_{C}=\frac{\left\{\left[y\;-\;\left(1\;-\;\frac{x}{2}\right)\right]\cdot M_{C}\right\}/\rho C}{\left\{\left[y\;-\;\left(1\;-\;\frac{x}{2}\right)\right]\cdot M_{C}\right\}/\rho C+\left(\left\{\frac{x}{2}\cdot M_{{\mathrm{SiO}}_{2}}\right\}/{\rho \mathrm{SiO}}_{2}\right)+\left(\left\{1\;-\;\frac{x}{2}\;-\;\frac{3z}{4}\cdot M_{\mathrm{SiC}}\right\}/\rho \mathrm{SiC}\right)}$$(with M being the molar mass and ρ the density of the constituent phases in the studied silicon oxycarbides), may be adjusted between 0 and ca. 58 vol.% (Table 2). Clearly, the synthesis of silicon oxycarbides from suitable precursors allows one to tailor their amount of carbon, especially that of the sp^2^-hybridized, segregated carbon. Consequently, it is expected that the charge carrier transport behavior in those silicon oxycarbide-based materials may be gradually set from that of an insulator to that of a semimetal, as shown previously [1,16,34].

The prepared silicon oxycarbides were investigated by means of Raman spectroscopy. Figure 1 displays the visible Raman spectra (λ = 514 nm) of the sample C-11, which was heat-treated in argon at different temperatures from 1000 to 1800 °C. The Raman spectrum of the sample prepared at 1000 °C exhibited two broad signals at ca. 1330 cm^−1^ and ca. 1610 cm^−1^ representing the D and G modes, along with a broad, low-intensity signal at 1120 cm^−1^, the so-called T band [45]. No significant change is observed in the Raman spectra of the C-11 samples heated at temperatures up to 1400 °C. However, in the spectrum of the sample annealed at 1400 °C, the T band, related to residual sp^3^ carbon, nearly vanished and the full width at half maximum (FWHM) of the D band significantly narrowed along with an increasing intensity. By comparing the spectra of the samples heat treated at 1400 and 1600 °C, it could be concluded that the width and intensity of the G band did not significantly change, though a small shoulder around 1620 cm^−1^ arose. This signal was assigned to the D’ band [45,46,47] and indicate the evolution of small nano-graphitic clusters in the silicon oxycarbide sample. This has been substantiated by the appearance and development of two signals (2D and D + G) in the second-order spectrum.

Thus, it can be stated that the segregated carbon phase in C-11 increased its ordering as the temperature of heat treatment increased. With increasing crystalline order of the carbon phase, the D band continued to narrow and the D’ band became more and more pronounced together with a shift of the G band towards lower wavelengths (Figure 1). Although the advancing graphitization did not lead to an increase of the intensity of the G band, the two signals in the second-order range of the spectrum clearly improved their intensity. In the sample annealed at 1800 °C, the intensity of the G band was notably increased and the wavelength was downshifted by 30 cm^−1^, with the D’ band remaining as a small shoulder. In addition, the second-order spectrum markedly differed from those of the samples annealed at lower temperatures, as the band at 2700 cm^−1^ had apparently grown in intensity and was displaced by 15 cm^−1^ to the lower wavelength. According to Pimenta et al. the shape and shift of this band resembled turbostratic stacking of several graphene sheets along the c-axis, as the width of the band (w = 77 cm^−1^) was significantly larger than that of monolayer graphene (w = 24 cm^−1^) [47]. Thus, the progressive changes in the Raman spectra of C-11 might be correlated to the evolution of the segregated carbon from amorphous to nanocrystalline and eventually to turbostratic microcrystalline carbon.

Figure 2 shows the lateral crystal size L_a_ and its progress with increasing the annealing temperature for the C-11 based samples. As shown by Tuinstra and König, the relative intensity ratio I_D_/I_G_ of the D and G bands depends inversely on L_a_ for graphitic samples; whereas, L_a_ has been best described for highly disordered carbons upon using the ratio of the integrated areas A_D_/A_G_ of the D and G band instead of their intensities [48,49]. Consequently, the disordered carbon phase in SiOC/C was assessed by using the following equation [49,50].
(2)$$L_{a}=\left(2.4\cdot 10^{-10}\right)\cdot \lambda_{l}^{4}\cdot {\left(\frac{A_{D}}{A_{G}}\right)}^{-1}$$
with λ_l_ being the laser wavelength. As clearly indicated in Figure 2, the lateral crystal size L_a_ was not affected by the treatment temperature in the temperature range from 1000 to 1500 °C and only slightly varied around a value of ca. 7.5 nm. This is in agreement with the work of Takai et al. who obtained similar values (L_a_ = 5–7 nm) for amorphous carbon films heat treated at temperatures up to 1500 °C. At higher temperatures, the crystalline lattice started to grow, as intrinsic lattice defects were removed. The onset of graphitization was emphasized by the increase in intensity of the G band and the clear development of D’ at temperatures beyond 1600 °C, as shown in Figure 1. Thus, the L_a_ value found in the C-11 sample annealed at 1800 °C was ca. 20 nm. This is in excellent agreement with a study of Cançado et al. who investigated the graphitization behavior of diamond-like carbon thin films at temperatures ranging from 1800 to 2700 °C, which revealed a L_a_ value of 20 nm for the films annealed at 1800 °C [50] (see empty symbols in the inset of Figure 2).

As the series of SiOC/C samples in the present study showed, the content of the segregated carbon phase in the prepared materials can be adjusted in a broad range. Within this context, Raman spectroscopic investigations were done in order to rationalize whether the amount of segregated carbon in SiOC/C ceramic might have an effect on the lateral crystal size L_a_. Figure 3 plots the L_a_ as a function of the volume fraction of the segregated carbon phase in the studied SiOC/C ceramics and revealed a constant L_a_ value of 9 ± 1 nm for samples showing carbon contents above ϕ_C_ > 10 vol.%; whereas the samples possessing of lower carbon contents, (ϕ_C_ < 10 vol.%) exhibited smaller L_a_ values of 6 ± 2 nm. Thus, it may be concluded that the large(r) amount of segregated carbon in silicon oxycarbides correlates to its relative higher graphitization degree [39].

The evolution of the segregated carbon phase in silicon oxycarbides as a function of the annealing temperatures was also assessed by considering the interdefect distance L_D_. L_D_ was calculated according to Ref. [51] (Equation (3), A_D_ and A_G_ represent the areas of the D and G band, respectively).
(3)$$L_{D}^{2}=1.8\;\cdot 10^{-9}\cdot \lambda_{L}^{4}\cdot {\left(\frac{A_{D}}{A_{G}}\right)}^{-1}$$

In Figure 4, the A_D_/A_G_ ratio in C-11 samples annealed at different temperatures was plotted as a function of the L_D_ value. It is clearly shown that A_D_/A_G_ decreased and L_D_ increased as the annealing temperature of C-11 increased from 1000 to 1800 °C. In the C-11 sample prepared at 1000 °C, a L_D_ value of 6 nm was obtained, which means that at least one defect is present within the average lateral size of the crystallite (L_a_ (1000 °C) = 7.5 nm). By increasing the annealing temperature up to T = 1400 °C, L_D_ increased giving values in the range of L_D_ = 8–10 nm. This correlated to the appearance end evolution of the D’ band in the Raman spectra of the corresponding samples and was similar to the behavior and evolution of nano-graphitic structures. Upon annealing at higher temperatures, the concentration/number of the intrinsic defects was further reduced, giving rise to larger L_D_ and L_a_ values. This is an expected trend, which agreed very well with the description of Ferrari et al. who identified two stages concerning the evolution of the I_D_/I_G_ ratio: the first stage corresponded to a high defect concentration situation in the carbon phase and there the I_D_/I_G_ ratio scales with L_a_ cf. I_D_/I_G_ = 1/L_a_^2^; whereas the stage two corresponded to a low defect concentration in carbon, with I_D_/I_G_ = 1/L_a_ [45]. Considering the proposed L_a_ value of ca. 2 nm as a border between the first and second stage and the values obtained for the studied C-11 samples, it could be concluded that the carbon phase present in the C-11 samples could be assigned to the second stage, independent of the annealing temperature.

Interestingly, not only the trend but also the values of A_D_/A_G_ ratio and L_D_ were comparable with graphene-based materials in which the disorder was induced and adjusted upon Ar^+^ ion bombardment [52] (empty circles in Figure 4).

The significant effect of the annealing temperature on the ordering and defect concentration in the sp^2^ carbon phase of SiOC/C materials studied in the present work might obviously correlate to their electronic transport. This aspect was addressed by investigating silicon oxycarbide ceramics possessing various amounts of the segregated carbon phase and annealed at different temperatures via dc conductivity and Hall measurements and will be discussed in the following.

Figure 5 shows the (logarithmic) room temperature dc conductivity of the SiOC/C samples synthesized at temperatures in the range from 1000 to 1600 °C as a function of the content of sp^2^-hybridized segregated carbon. It is clearly seen that the dc conductivity shows a dependence on the sp^2^ carbon content in SiOC/C, which may be correlated to percolative systems. The critical concentration ϕ_cri._ (percolation threshold) of sp^2^ carbon is shown to depend on the annealing temperature and varied from ca. 20 vol.% for the sample series prepared at 1000 °C to ca. 6 vol.% for the samples annealed at 1600 °C (Figure 5). It is tempting to attribute the decrease of the percolation threshold to an increase in the aspect ratio of the segregated carbon phase, as the threshold typically depends on the shape of the conductive filler in many composite materials [53,54,55]. For instance, Cordelair et al. computed in a case study the insulator–conductor transition in SiOC/C-based materials, showing that the percolation threshold decreases from 20 to 5 vol.% as the shape of the conductive filler is changed from spherical to rod-like [35]. However, no reliable information on the aspect ratio can be obtained for SiOC/C, due to the random 3D carbon morphology and the lack of suitable characterization methods.

The observed temperature-dependent shift for ϕ_cri._ in the SiOC/C ceramics may therefore be also attributed to changes within the carbon network itself, e.g., the decrease of defects concentration and/or better alignment of carbon sheets allowing enhanced electrical transport in conjugated bonds. This possibility has been addressed upon assessing the critical exponent *t* in the studied SiOC/C samples cf. $\sigma_{C}=\sigma_{0}\cdot {\left(\phi \;-{\;\phi }_{cri.}\right)}^{t}$, where σ_0_ is a proportionality constant, σ_C_ the conductivity of the composite, ϕ_cri._ the critical concentration below the composite that is insulating (percolation threshold) and *t* the critical exponent. The value of the critical exponent *t* is known to depend on the lattice dimensionality and was calculated to possess a value of ca. 2.0 in three-dimensional lattices/systems [56,57]. Deviations from this behavior have been reported for many different composites [58]. According to Balberg, these deviations can be assigned to tunneling-percolation processes with strong fluctuations in tunneling distances thus causing high values for *t*, even up to 6.0 [59,60].

Non-universal behavior is frequently observed in dispersed composites, whereas universal behavior is seen more often in clustered samples [58]. The non-universal behavior may result from tunneling/hopping processes between two adjacent conductive particles [61], e.g., taking quantum mechanics into account the critical exponent becomes dependent on the mean tunneling distance and in principle does not obey any upper limit.

As clearly shown in Figure 6, the value of *t* in SiOC/C sample series prepared at 1600 °C (with a percolation threshold at 6 vol.%) was ca. 4.3 and thus these materials should be considered as being significantly different than three-dimensional systems, for which the geometry and aspect ratio of the conductive phase mainly determined their behavior. The *t* value obtained for the SiOC/C ceramics in the present study were similar to those obtained, e.g., for RuO_2_/silica composites (*t* = 3.8) [58].

Only sparse information about the conduction mechanism in PDCs is available in the literature. Haluschka and Engel reported on the conduction behavior of SiCN/C (ϕ_C_ = 5 vol.%) and SiOC/C (ϕ_C_ = 12 vol.%) prepared at T = 1500 °C and T = 1300 °C, respectively [36,66]. Both authors interpreted their data considering the variable range hopping model (VRH) postulated by Mott and Davis [67,68]. Haluschka et al. showed furthermore that the conduction occurs by phonon-assisted hopping near the Fermi-energy and not near the mobility edges [36]. Engel estimated the density of states near the Fermi energy to be N = 10^17^ − 10^19^ eV^−1^ cm^3^, by assuming a localization length of α^−1^ = 10 Å, which is typical for amorphous semiconductors [66]. Based on the correlation between the parameters σ_0_ and T_0_ in the general equation for transport in localized states (i.e., $\sigma =\sigma_{0}\cdot \left[-{\left(\frac{T_{0}}{T}\right)}^{1/\beta }\right]$, Ma et al. investigated a SiCN/C system (ϕ_C_ = 22 vol.%) considering the nearest neighbor hopping, also called band tail hopping (BTH), as the main transport mechanism [63].

Figure 7 shows the Arrhenius plots for the electrical conductivity of three C-11 samples prepared at 1000 °C, 1400 °C and 1600 °C. A linear dependency of σ with T^−1^ is evident for all samples, indicating an activated transport and activation energies E_a_ of 0.4 eV, 0.23 eV and 0.04 eV for the samples prepared at 1000 °C, 1400 °C and 1600 °C, respectively. For the sake of comparison, a SiCN/C system prepared at 1300 °C showed an E_a_ value of 0.15 eV [36] and was thus similar to the C-11 sample prepared at 1400 °C.

Plotting the conductivity data against T^−1/β^ (β = 3,4) yields no clear evidence of a non-Arrhenius behavior, underlining the presence of thermally activated transport in all the investigated samples. However, one cannot rule out that both VRH and thermally activated transport exist in this temperature range, thus further measurements are needed for an unambiguous mechanistic clarification. The decreasing activation energies with increasing the synthesis temperature indicate a continuous reduction of the band gap in the SiOC/C system and may be related to the evolution of the carbon phase with the temperature. Interestingly, the activation energy of the C-11 ceramic sample annealed at 1600 °C, i.e., 0.04 eV, was comparable to that of glassy carbon (0.03 eV [69]).

Figure 8 shows the activation energy E_a_ as a function of the segregated carbon in the SiOC/C samples prepared at 1600 °C. Three different electrical transport regimes can be distinguished, with activation energies decreasing from ca. 1 eV in regime I to ca. 0.3 eV in regime II and further to ca. 0.03 eV in regime III.

In regime I (Figure 8), conduction is considered to occur within the amorphous silica network, as the amount of sp^2^ carbon is very low (ϕ_C_ < 1 vol.%) [70,71]. According to Hapert, high concentration of silicon dangling bonds is present within amorphous silica SiO_x_, showing a VRH transport within localized states [70,71]. A VRH conduction mechanism is also likely for low-carbon SiOC/C materials.

In regime II (ϕ_C_ = 2–6 vol.%, Figure 8), the sp^2^-hybridized carbon precipitations were still separated by the SiOC matrix but the resulting potential barrier for the transport was lowered (E_a_ ≈ 0.3 eV) as compared to that of regime I. The low activation energy arose from additional carbon defects states within the band gap presumably overlapping with the silicon dangling bonds states. A very similar activation energy (E_a_ = 0.3 eV) has been found in carbon-based materials by Dasgupta et al., who investigated hopping transport between graphitic islands separated by a lower conducting sp^3^ carbon layer [72].

In regime III (Figure 8), the carbon concentration was above the percolation threshold for samples prepared at 1600 °C. Thus, the electrical transport was considered to be dominated by the continuous, three-dimensional carbon sp^2^ network, yielding activation energies in the range of E_a_ from 0.035 to 0.02 eV, which is in excellent agreement with the conduction energies found in glassy carbon [73]. The transition from semiconducting to semi-metallic conduction is accompanied by the reduction of the gap between carbon π and π* band states. Carbon essentially controlled the density of states, whereas the SiOC states were of minor importance within regime III (Figure 8). Hence, the same transport mechanism (i.e., as in regime III, Figure 8) was assumed to prevail for all samples above the percolation threshold.

The electrical behavior of the SiOC/C samples investigated in the present study could be correlated with their Raman data. Upon assuming that the width of the band gap depends on the concentration of carbon precipitations cf. $E_{a}=\frac{2.1}{L_{a}}$ for carbon-based semiconductors [74] and $E_{a}=\frac{7.7}{L_{a}}$ for graphitic-like materials [75], the activation energy E_a_ may be estimated with the help of the L_a_ value. Thus, an activation energy of E_a_ = 0.28 eV was calculated for semiconducting samples with an average lateral crystal size of L_a_ ca. 7.5 nm, which was in excellent agreement with the E_a_ values obtained from the conductivity measurements (see Figure 8, regime II); whereas for materials exhibiting a value of 9 nm for L_a_, an activation energy of E_a_ = 0.085 eV was obtained, which was though larger than the E_a_ determined in regime III (E_a_ = 0.035–0.02 eV, Figure 8).

Information about the density and mobility of the dominant charge carrier type in the SiOC/C ceramic materials was obtained from measurements of the Hall effect. Figure 9 exhibited the carrier density (N) and mobility (µ) as a function of the carbon content for SiOC/C samples prepared at 1600 °C. Literature values of an SiOC/C synthesized at T = 1550 °C are displayed for comparison [76].

The charge carrier density/mobility can be furthermore estimated from conductivity measurements if the electrical transport solely depends on one charge carrier type (electrons or holes), as follows: σ=e·N·μ. The carrier densities derived from dc measurements on SiOC/C samples are shown in Figure 9 (green circles) as a function of the carbon concentration; the charge carrier densities were calculated using the experimentally determined mobility of µ = 3 cm^2^/Vs, which is similar to that of turbostratic carbon (µ = 3.1–6.3 cm^2^/Vs [77,78]). For SiOC/C samples possessing carbon contents higher than ca. 8 vol.%, the charge carrier densities derived from the dc conductivity data showed an excellent agreement to those obtained from the Hall effect measurements and were furthermore comparable to values determined for glassy carbon [79,80]. However, SiOC/C samples with carbon contents significantly lower than 8 vol.% showed strong discrepancy between the dc-conductivity-derived data and those obtained from the Hall effect measurements. Unfortunately, Hall measurements for samples with rather low conductivity or electrical non-uniformity are experimentally challenging and consequently the few data obtained for the SiOC/C samples with low contents of segregated carbon (i.e., <8 vol%) do not allow any clear statement.

Figure 10 plots the Hall coefficient (R_H_ = 1/Ne = µ/σ) of SiOC/C samples as a function of their sp^2^-hybridized carbon content. It is shown that SiOC/C samples possessing carbon content in the range from 8 to 43 vol% show a Hall coefficient of 5 × 10^−2^ cm^3^/C, which does not depend on the amount of carbon in SiOC/C and is similar to that of glassy carbon (2.8 × 10^−2^ cm^3^/C) [77]. The positive sign of the Hall coefficient indicates p-type carrier transport. This may be related to vacancies (missing carbon atoms within the lattice of sp^2^ carbon) or to the removal of dangling bonds creating holes in the filled π-valence band [81].

Positive Hall coefficients comparable to those measured for SiOC/C samples in the present study were also reported for (nano)crystalline graphite [82,83,84]. It is shown that upon decreasing the sp^2^ carbon content in the studied SiOC/C samples (i.e., ϕ_C_ < 8 vol%), the Hall coefficient strongly increased, indicating less ordering of the carbon phase in SiOC/C materials containing low amounts of sp^2^ carbon. This result is supported by the Raman data showing a change of the lateral crystal size L_a_ and the average defect distance L_D_ in the same concentration region. Furthermore, the activation energy for the conductivity is shown to increase from 0.03 to 0.3 eV as the amount of free carbon decreased below 8 vol.%, as discussed above.

## 3. Materials and Methods

### 3.1. Materials Synthesis

The sol–gel route used in the present work to synthesize SiOC samples with varying carbon content was already reported in detail in [41,42,43]. Within this work, a series of additional samples was prepared according to the same general procedure using different alkoxysilane precursors. An exemplary description of the synthesis procedure is given in the following: 8.21 g triethoxysilane (0.05 mol) and 9.51 g triethoxyvinylsilane (0.05 mol) were mixed and stirred for 15 min. Subsequently, water (5.4 g, 3 mol) with a pH of 5 was added dropwise and the solution was stirred overnight to obtain a homogeneous clear solution. The solution was aged in air for 24 h without using any catalyst to obtain the corresponding gel, which was dried at 80 °C for 24 h.

Both commercially and synthesized precursors were thermally crosslinked in an alumina tube furnace (HTSS 810/10, Carbolite Gero, Neuhausen, Germany) at T = 250 °C for 2 h and subsequently pyrolyzed at T = 1100 °C for 2 h with a heating and cooling rate of 100 °C/h under a constant flow of argon (5 L/h). To ensure an oxygen-free atmosphere during thermal treatment, the chamber was evacuated three times and purged with high purity argon. The resulting black glassy powders were ground and sieved to a particle size ≤40 µm. The sieved powders (m = 1–2 g) were sealed within graphite foils and hot-pressed (Spark Plasma Sinter System, SPS-211Lx, Fuji Electronics Industrial Co. Ltd., Technical, Fujimi, Saitama, Japan at T = 1600 °C with a uniaxial load of P = 50 MPa for 15 min under high purity argon atmosphere and using a heating rate of 320 °C/min. Subsequently, the obtained dense monoliths were polished using a grinding machine (ZB 42T, Ziersch and Baltrusch, Ilmenau, Germany) equipped with a diamond grinding wheel to ensure a defined geometrical shape with plane surfaces.

### 3.2. Materials Characterization

The elemental composition of the as-prepared silicon oxycarbide samples was determined by hot gas extraction. The carbon content of the samples was measured with a carbon analyzer (CS 800, Eltra GmbH, Neuss, Germany), which detects and quantifies the oxidized carbon species by means of IR spectroscopy. An N/O analyzer (Leco TC-436, Leco Corporation, St. Joseph, MI, US) was used to determine the oxygen content. The silicon weight fraction was considered to be the difference to 100 wt %, assuming no other elements being present in the sample.

Visible Raman spectra were recorded with a Horiba HR800 micro-Raman spectrometer (Horiba Jobin Yvon GmbH, Bensheim, Germany) equipped with an Argon laser (λ = 514.5 nm). The excitation line has its own interference filter (to filter out the plasma emission) and a Raman notch filter (for laser light rejection). The measurements were performed with a grating of 600 and a confocal microscope (magnification 50×, NA = 0.5) with a 100 μm aperture, giving a resolution of approximately 2–4 µm. The laser power (20 mW) was attenuated by using neutral density filters; thus, the power on the sample was in the range from 6 µW to 2 mW. All spectra were background subtracted, smoothed (SMA, simple moving average) and fitted to Lorentzian line shapes using Origin Pro 9.1.0G.

For the temperature-dependent measurements of the electrical conductivity, cylindrical samples were clamped between two electrodes (two-point method) within a quartz tube and subsequently hermetically sealed before the tube was introduced into a cylindrical alumina furnace (LOBA 1400-45-400-1, HTM Reetz GmbH, Berlin, Germany). Temperatures up to T = 800 °C were available and were verified by a thermocouple placed next to the sample. The quartz tube was sealed and then evacuated and flushed three times with argon (purity ≥ 99.5%, Air Liquide, Frankfurt, Germany) before the heating program was started. Impedance spectroscopy measurements were performed using an alpha-A high performance modular measurement system (Novocontrol Technologies, Montabaur, Germany). The measurements were conducted in a frequency range of ν = 0.1–3 MHz at an applied voltage of U = 0.1 V. No sign of a second contribution to the conductivity (contact or matrix dependent resistance) was observed in the most extensively investigated samples. Since the bulk resistance obtained from the impedance spectra equaled the dc resistance, only dc data were reported in the following.

Hall measurements were conducted to measure the electrical conductivity, the Hall constant, and the carrier mobility and carrier density of the samples using a system built at the Fraunhofer Institute for Physical Measurement Techniques in Freiburg, Germany (IMP-HT-Hall-900K). Square-shaped samples (1 mm × 10 mm × 10 mm) were placed into a sample holder installed within a vacuum chamber. The chamber was subsequently evacuated and flushed with argon. The samples were contacted according to the van der Pauw measurement theory [85]. Thus, four points were connected at the edge of the sample whereas on one side of the sample the current was applied and on the other side the resulting voltage was measured. The separation of the current and voltage electrode eliminated the contact and led resistance from the measurement and consequently increased the accuracy. The measurement was done in all four permutations of the contacts, and due to the used AC-resistance bridge in both polarizations. The vacuum chamber was moved into the positive and negative field position to determine the Hall coefficient and the carrier concentration and mobility.

## 4. Conclusions

The results obtained in the present work indicate that the charge carrier transport behavior of SiOC/C materials correlated with the amount and the ordering of their sp^2^-hybridized free carbon phase, which could be tuned upon adjusting the molecular structure of the precursor and the pyrolysis temperature, respectively. Thus, the sp^2^ free carbon content could be varied within a broad range from nearly negligible fractions to as high as ca. 58 vol.%. Whereas the ordering of the free carbon phase could be significantly improved by increasing the treatment temperature. The percolation threshold in SiOC/C was obviously a function of the ordering of the carbon phase and could be significantly reduced from ca. 20 vol.% (as for samples prepared at 1100 °C) to ca. 6 vol.% for samples exposed to 1800 °C. SiOC/C shows near the percolation threshold a charge carrier transport dominated by tunneling and hopping processes between the carbon-based precipitates; whereas VRH transport in localized states of the glassy SiOC matrix and carbon-dominated p-type transport is concluded for carbon contents being far below or beyond the percolation threshold in SiOC/C, respectively.

## Figures and Tables

**Figure 1 molecules-25-05919-f001:**
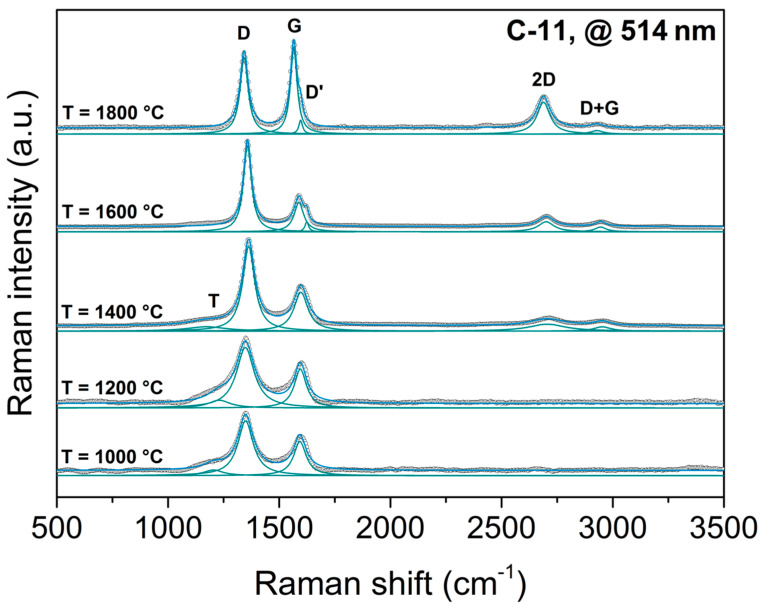
Raman spectra of sample C-11 heat-treated at different temperatures in the range from 1000 to 1800 °C (laser wavelength was 514 nm). Green lines represent deconvoluted bands, black circles indicate the experimental curve and the blue line is the fitted curve.

**Figure 2 molecules-25-05919-f002:**
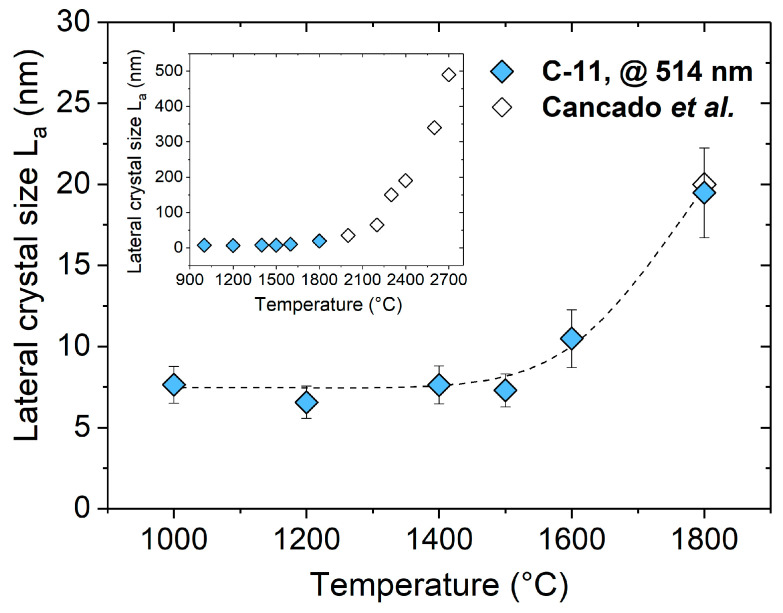
Evolution of the lateral crystal size L_a_ in C-11 as a function of the treatment temperature. The open data points in the inset represent the work of Ref. [50], indicating the evolution of L_a_ with the annealing temperature for diamond-like thin films.

**Figure 3 molecules-25-05919-f003:**
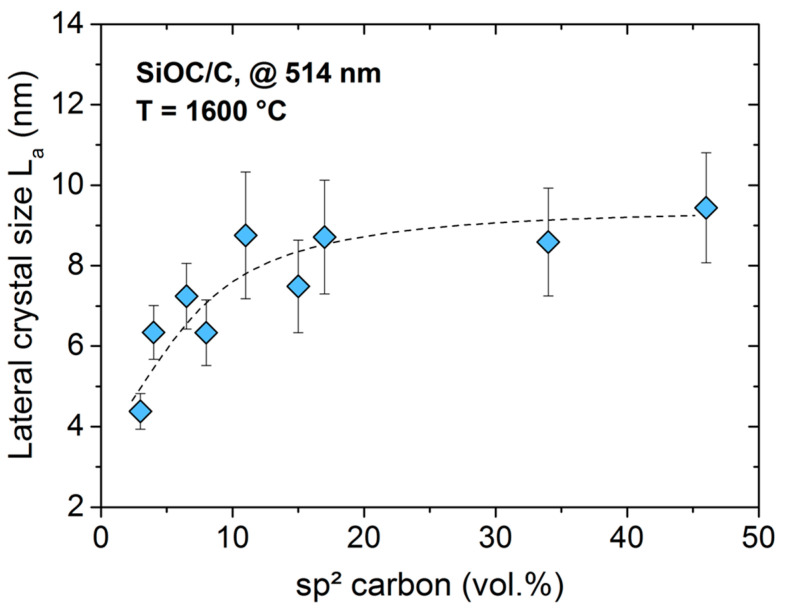
Plot of the lateral crystal size L_a_ (blue diamonds) as a function of the amount of carbon. Bars represent standard deviation.

**Figure 4 molecules-25-05919-f004:**
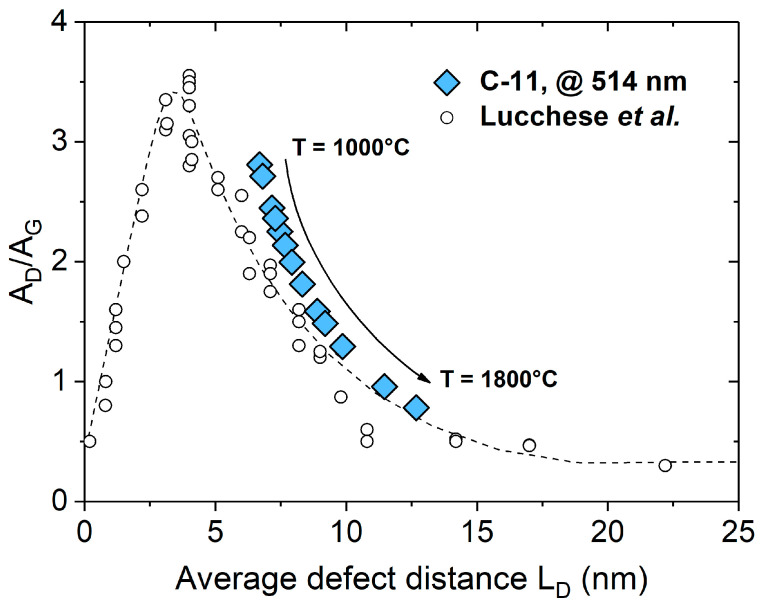
Evolution of the average distance between defects L_D_ within the carbon of SiOC/C synthesized at different temperatures. The empty symbols correspond to a graphene-based material in which the disorder and defect concentration were altered upon argon ion bombardment [52].

**Figure 5 molecules-25-05919-f005:**
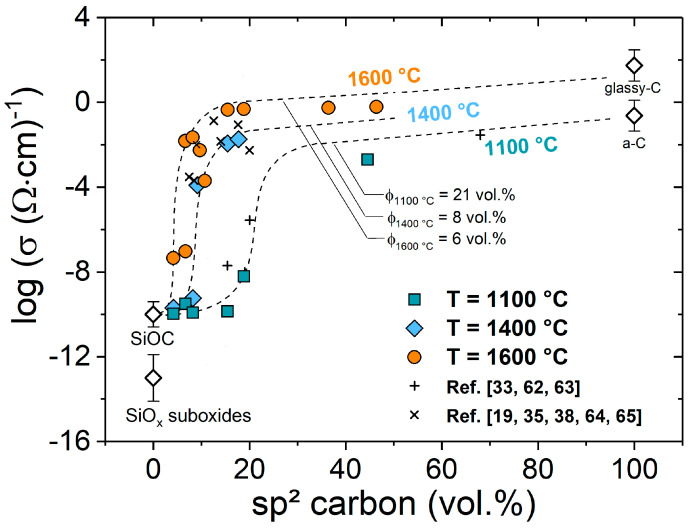
The electrical conductivity of SiOC/C as a function of the sp² carbon for three different temperature series (a-C: amorphous carbon). The data were obtained from two-point conductivity measurements. Literature values [19,33,35,38,62,63,64,65] are added for the sake of comparison.

**Figure 6 molecules-25-05919-f006:**
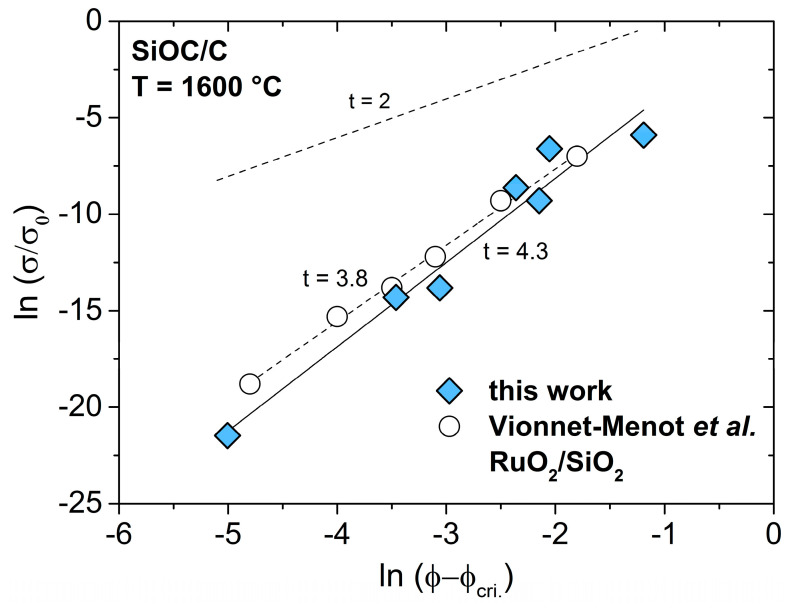
Electrical conductivity of SiOC/C materials as a function of the sp^2^ carbon concentration. Table 1. The upper dashed line represents the universal behavior with a slope of *t* = 2. The empty circles represent the electrical conductivity of piezoresistive RuO_2_/silica composites with varying RuO_2_ content [58].

**Figure 7 molecules-25-05919-f007:**
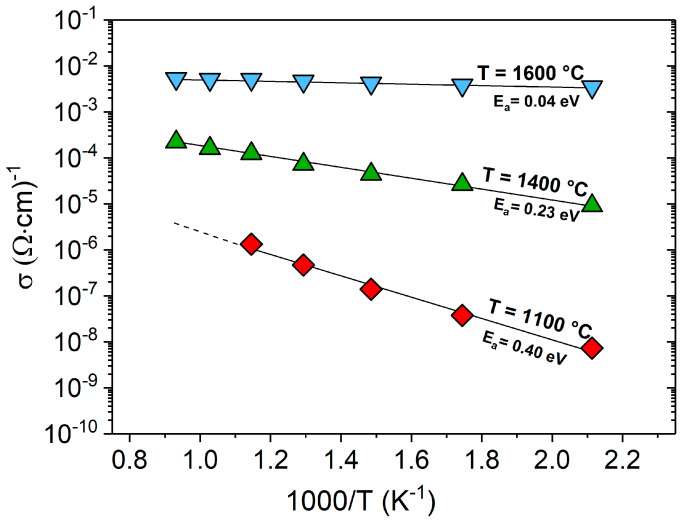
Electrical conductivity of C-11 as a function of the inverse of the treatment temperature.

**Figure 8 molecules-25-05919-f008:**
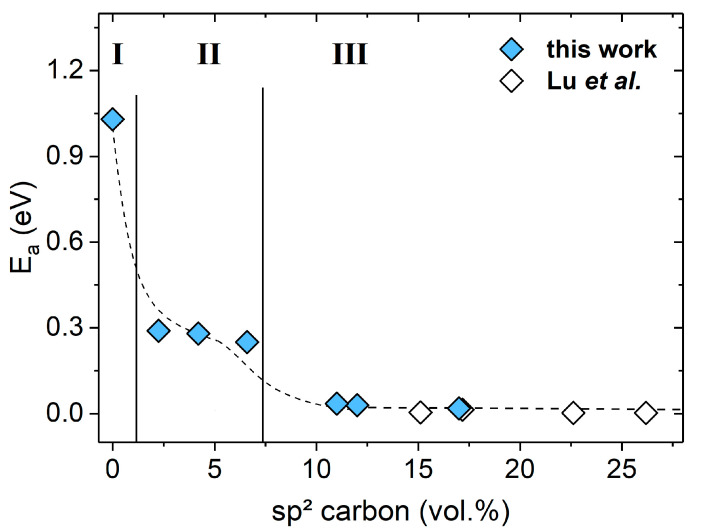
Plot of E_a_ as a function of the sp^2^ carbon content in SiOC/C, the empty symbols represents data published by Lu et al. [34]. The dotted line is shown only as a guideline, it does not represent any fit of the data. I, II and III represent the three conduction regimes as described in text above.

**Figure 9 molecules-25-05919-f009:**
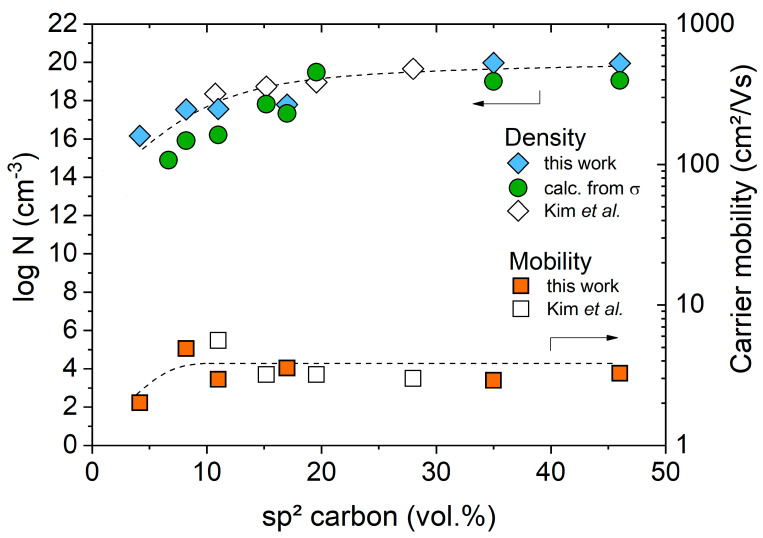
Charge carrier density (N) and mobility (μ) in SiOC/C materials prepared at T = 1600 °C; the empty symbols represent data from a case study published by Kim et al. [76].

**Figure 10 molecules-25-05919-f010:**
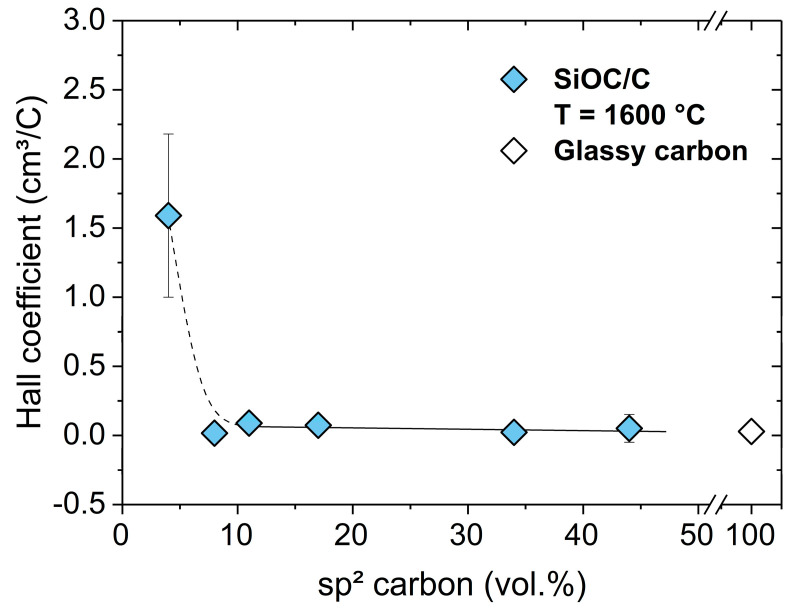
Plot of the Hall coefficient in SiOC/C samples synthesized at 1600 °C as a function of their sp^2^-hybridized carbon content. Glassy carbon is depicted as reference [75].

**Table 1 molecules-25-05919-t001:** SiOC/C-based samples considered in the present study. The number given in the sample name indicates the volume fraction of the segregated carbon phase.

	Sample	Precursors	Ceramic Yield (wt %)	Ref.
Sol-Gel Precursors		*TREOS:MDES molar ratio*		
	C-0	10:1	87.1	[41,42,43]
	C-1	2:1	88.4	
	C-2	1:2	86.6	
		*TREOS:TEMS molar ratio*		
	C-4	2:1	82.2	this work
	C-6	1:1	80.7	
	C-8	1:2	78.5	
		*TREOS:TEVS molar ratio*		
	C-13	1:1	85.0	this work
	C-15	1:2	87.6	
	C-18	1:3	88.2	
	C-20	1:4	85.1	
Preceramic Polymers	C-11	*PMS MK*	81.0	[44]
	C-17	*SPR 212*	85.6	[12]
	C-36	*Silres 604*	84.7	this work
	C-46	*SPR 688*	85.8	[12]
	C-60	*SPR 684*	82.0	[12]

**Table 2 molecules-25-05919-t002:** Elemental composition, empirical formula and estimated phase composition of selected SiOC/C samples.

Sample	Element Content (wt %)			Empirical Formula SiO_2(1 − x)_C_x_ + y	Volume Fraction (vol.%)		
	Si	O	C		SiO_2_	SiC	C
C-0	51.78	40.88	7.03	SiO_1.39_C_0.31_	83.2	16.8	-
C-1	50.97	42.14	6.89	SiO_1.45_C_0.28_ + 0.04C	84.27	14.55	1.18
C-8	48.38	38.69	12.93	SiO_1.42_C_0.29_ + 0.31C	77.42	14.38	8.20
C-11	46.65	39.40	13.95	SiO_1.48_C_0.26_ + 0.44C	76.75	12.24	11.01
C-17	49.13	26.44	24.43	SiO_0.94_C_0.53_ + 0.63C	54.49	27.83	17.68
C-36	36.53	27.90	35.57	SiO_1.34_C_0.33_ + 1.95C	51.91	11.66	36.43
C-46	13.72	38.23	48.05	SiO_0.63_C_0.69_ + 2.25C	26.86	26.71	46.42
C-60	27.96	16.93	55.11	SiO_1.06_C_0.47_ + 4.13C	30.19	12.28	57.53

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
