# Peer review of "Effect of the Content and Ordering of the sp2 Free Carbon Phase on the Charge Carrier Transport in Polymer-Derived Silicon Oxycarbides"

_molecules, 2020, doi:10.3390/molecules25245919_

Round 1

Reviewer 1 Report

The text is generally well organized and mostly clear with minor corrections listed below. The presentation of data for the wide range of carbon concentrations is somewhat limited, focusing on C-11 samples, at moderate concentration levels but demonstrating the effect of temperature to development of sp2 carbon structures. Broader effects of concentration are presented in Figs. 3, 5, 6 and 8-10. Connection is made to the assembly of sp2 carbon into integrated large scale conductive structures that may resemble glassy carbon in an SiOx and SiC matrix. No description of the optical density or other optical measurements were made.

Please see details comments below.

Line 32, correct “sp2 carbon content in the range from 1014 to 1019 cm-³” with superscript “2”, “14”, “19”, and minus sign “-3”.

Line 39, “the fact that they exhibit and combine in a unique manner…” is unclear since the uniqueness is not yet defined. It makes more sense as “the unique way in which they exhibit and combine….”

Line 68, correct “ca. 1000-1200 °C” to “ca. 1000 °C to 1200 °C.” The same for lie 117, line 123, line 155, line 163, line 187, line 215, line 271.

Line 80, correct “107 to 1010 Ω/m” to “107 Ω/m to 1010 Ω/m.” Units should be given for each value (also line 190, line 357, etc).

Line 86, “the free carbon barely contributes to…”

Line 99, This may be unclear. Please define as “from ca. 7 wt.% (sample C-0) up to 55 wt.% (sample C-60).”

Line 106, consider placing the equation on a separate line; correct rho in “and ρ the density” to appear as in the equation.

Line 109, Superscript in “sp2-hybridized,”

Line 163, Throughout, subscripts are not placed below the text in terms like “La” and “{phi}C.”.

Line 166, Should be “Figure 2.”

Line 174, change “value of 9 ± 1 nm” to “value of (9 ± 1) nm.” Also in line 176.

Line 177, it is not clear what type of ordering is meant in “correlates to its relative higher ordering.”

Figure 5, Please define “a-C” or write out as “amorphous”. Also the method used to derive the dotted curves should be described.

Line 257, “few information” should be corrected to “sparse information”.

Line 296, rearrange as “in carbon-based materials has been found by Dasgupta et al.,” for clarity.

Figure 8, The dotted line may be disputed and should be removed unless it is described by rigorous analysis.

Figure 9, if the dotted lines are guides to the eye only, they should not extend to lower values where there is not data. This applies also to Figure 10.

Table 2 appears before Table 1. These labels should be exchanged.

Line 414, What tube is referred to in “The tube was evacuated and flushed… before the heating program was started.” If this is the sample tube, shouldn’t this be changed to “… before sealing.”?

Author Response

Thank you for the valuable Reviewers comments. We have seriously considered the Reviewers comments and revised our manuscript with respect to their suggestions. All changes performed in the text are highlighted.

  • The text is generally well organized and mostly clear with minor corrections listed below. The presentation of data for the wide range of carbon concentrations is somewhat limited, focusing on C-11 samples, at moderate concentration levels but demonstrating the effect of temperature to development of sp2 carbon structures. Broader effects of concentration are presented in Figs. 3, 5, 6 and 8-10. Connection is made to the assembly of sp2 carbon into integrated large-scale conductive structures that may resemble glassy carbon in an SiOx and SiC matrix. No description of the optical density or other optical measurements were made.

Thank you for the critical comments, which we considered while revising the manuscript. No optical measurements were made on the materials, as those are black due to the contained sp2 hybridized segregated carbon phase and thus strongly absorbing.

  • Line 32, correct “sp2 carbon content in the range from 1014 to 1019 cm-³” with superscript “2”, “14”, “19”, and minus sign “-3”.

This has been done accordingly.

  • Line 39, “the fact that they exhibit and combine in a unique manner…” is unclear since the uniqueness is not yet defined. It makes more sense as “the unique way in which they exhibit and combine….”

This has been corrected as suggested.

  • Line 68, correct “ca. 1000-1200 °C” to “ca. 1000 °C to 1200 °C.” The same for lie 117, line 123, line 155, line 163, line 187, line 215, line 271.

This has been done accordingly.

  • Line 80, correct “107 to 1010 Ω/m” to “107 Ω/m to 1010 Ω/m.” Units should be given for each value (also line 190, line 357, etc).

This has been done as suggested.

  • Line 86, “the free carbon barely contributes to…”

This has been corrected accordingly.

  • Line 99, This may be unclear. Please define as “from ca. 7 wt.% (sample C-0) up to 55 wt.% (sample C-60).”

This has been corrected accordingly.

  • Line 106, consider placing the equation on a separate line; correct rho in “and ρ the density” to appear as in the equation.

This has been corrected accordingly.

  • Line 109, Superscript in “sp2-hybridized,”

This has been corrected accordingly.

  • Line 163, Throughout, subscripts are not placed below the text in terms like “La” and “{phi}C.”.

This has been corrected accordingly.

  • Line 166, Should be “Figure 2.”

This has been corrected.

  • Line 174, change “value of 9 ± 1 nm” to “value of (9 ± 1) nm.” Also in line 176.

This has been done accordingly.

  • Line 177, it is not clear what type of ordering is meant in “correlates to its relative higher ordering.”

This has been clarified accordingly by using “higher graphitization degree” instead.

  • Figure 5, Please define “a-C” or write out as “amorphous”. Also, the method used to derive the dotted curves should be described.

This has been done accordingly.

  • Line 257, “few information” should be corrected to “sparse information”.

This has been done accordingly.

  • Line 296, rearrange as “in carbon-based materials has been found by Dasgupta et al.,” for clarity.

This has been rearranged as suggested.

  • Figure 8, The dotted line may be disputed and should be removed unless it is described by rigorous analysis.

We agree to the Reviewer. We mention in the revised manuscript in the Figure caption that the dotted line has only eye guideline purpose and does not represent any fit of the data.

  • Figure 9, if the dotted lines are guides to the eye only, they should not extend to lower values where there is not data. This applies also to Figure 10.

This has been corrected accordingly.

  • Table 2 appears before Table 1. These labels should be exchanged.

This has been corrected accordingly.

  • Line 414, What tube is referred to in “The tube was evacuated and flushed… before the heating program was started.” If this is the sample tube, shouldn’t this be changed to “… before sealing.”?

This has been clarified accordingly.

Reviewer 2 Report

COMMENTS TO Manuscript ID: molecules-1020671

Title: Effect of the Content and Ordering of the sp2 Free Carbon Phase on the Charge Carrier Transport in Polymer-Derived Silicon Oxycarbides

Journal: Molecules

This work deals with the correlation between the amount of carbon in Silicon Oxycarbide Glasses and the charge carrier transport. They carried out a nice analysis of the evolution of the sp2 carbon as a function of the temperature by means of Raman spectroscopy. On the other hand, they found different conduction regimes depending of the carbon content. Hall effect results indicated that these samples presented a p-type carrier.

Even the article is interesting a few corrections and clarifications have to be carried out in order to improve the scientific content of the article.

First of all, English is quite good but some errors in the text can be observed.

Page 5, Figure 4 is Figure 2

Page 6, line 175, Figure 6 is Figure 2

Page 10 line 289, authors mentioned Hapert’s work. However, such name does not exist in bibliography chapter.

Page 10, line 291, authors indicated ref 291. However, such reference does not exist.

Page 4, equation 1, which is the meaning of  lambda4?.

Please, check the text, figures, constants, etc., in order to correct all errors and then facilitate to review and the understanding of the paper.

Figure 1 presents the Raman spectra of the C-11 sample at temperatures between 1000ºC and 1800ºC. They have performed a spectra deconvolution. Which type of band are selected for deconvolution, Gaussian, Lorentzian, Voight, mixed, etc.?? Please, indicate it in the text.

On the other hand, when figure 1 is examined, baseline is below the experimental data. Then, is the baseline correctly subtracted?

Authors performed deconvolution over almost all Raman bands. However, the article entitled “Evolution of the Raman Spectrum with the Chemical Composition of Graphene Oxide”, J. Phys. Chem. C 2017, 121, 20489−20497 DOI: 10.1021/acs.jpcc.7b06236, presents a nice deconvolution that can be taken into account. At least, authors have to include the band located between D and G bands (D’’ band) in order to obtain a better fit of D and G bands.

In page 4, line 135, authors indicated that even the graphitization advances, the intensity of the G band does not. How do they know that the graphitization advances? Please, reference such affirmation. On the other hand, as suggestion, authors can carry out XRD analysis in order to check if the d parameter decreases or increases. Also, TEM analysis can help. Please, check the article entitled “Comparative XRD, Raman, and TEM Study on Graphitization of PBO-Derived Carbon Fibers” J. Phys. Chem. C 2012, 116, 257–268 DOI: /10.1021/jp2084499.

Figure 3 present data from La as a function of the amount of carbon. For example, for SiOC samples containing about 10% of sp2 carbon in vol%, La is about 8.8 nm. Error bars is about ±2 nm. In my opinion, it means that the error in the measurement is high. Does it mean that your sample is not homogeneous and depending of the analysis area you can obtain different results?

Page 6, line188, authors said: “…which means that at least one defect is present within the average lateral size of the crystallite…” Please, can you explain such affirmation? What do you mean “with at least one defect”?

Page 8, line 248, I think that t value for SiOC/C sample is 4.3 (not 3.8) as it can be seen in Figure 6.

Page 8, lines 250-251 authors compare the t results of SiOC/C with RuO2/silica composites. Which are the similarities between both materials? Do they have the same structure, etc.? Explain it, please,

Figure 5, in page 8, red dots blue rhomb and dark blue squares are from 1600, 1400 and 1100ºC heat treatments. Table 2 shows the SiOC/C selected samples. In such table there are 8 samples. However, in figure 5 you can find 10 red dots. Does it mean that some results related to electrical conductivity for 1600ºC samples are repeated?

Page 9, lines 279-280, authors said: “Interestingly, the activation energy of the C-11 ceramic sample….., i.e. 0.04 eV, is comparable to that of glassy carbon (0.03 eV)”. Why is it interesting? Can you explain? On the other hand, I found articles with different materials such PAN in which the activation energy is similar.

Page 10, line 288, authors indicated that the conduction occurs within the amorphous silica due to the low carbon concentration present in the SiOC matrix. However, they do not present any reference or if they do not find any, try to explain.

Page 11, authors mentioned that electrical behaviour of SiOC/C samples can be correlated with the concentration of carbon precipitations. They said that results obtained from La is in excellent agreement with values obtained from conductivity measurements. How do you calculate such result? From which plot? Which fitting? It is important to explain in order to facilitate the experimentation for other researchers.

It is very tentative to challenge to say that the activation energy in regime III calculated form sp2 carbon is in the same order as the obtained for other methods. From your method Ea= 0.085 eV and other methods Ea=0.035-0.02 eV. The difference is almost 2.5 times higher. In my opinion is a high difference.

In different plots, some data is presented with error bars, however other data do not have error bars. Why? Does it mean that some data are obtained with errors?

The Hall effect is the movement of charge carriers through a conductor towards a magnetic attraction. It means that such effect can be attributed to an electrically conductor material. I think that as the carbon content decreases, the conductivity also decreases. Then, if the material does not present any conductivity or is very low, the Hall effect would be difficult to detect or even it is hard to measure.

Then, the article is interesting but they have to be improved scientifically (maybe DRX, TEM, …) and make some clarifications.

Author Response

Thank you for the valuable Reviewers comments. We have seriously considered the Reviewers comments and revised our manuscript with respect to their suggestions. All changes performed in the text are highlighted.

  • This work deals with the correlation between the amount of carbon in Silicon Oxycarbide Glasses and the charge carrier transport. They carried out a nice analysis of the evolution of the sp2 carbon as a function of the temperature by means of Raman spectroscopy. On the other hand, they found different conduction regimes depending of the carbon content. Hall effect results indicated that these samples presented a p-type carrier. Even the article is interesting a few corrections and clarifications have to be carried out in order to improve the scientific content of the article.
  • First of all, English is quite good but some errors in the text can be observed.

We performed careful proof reading accordingly.

  • Page 5, Figure 4 is Figure 2

This has been corrected accordingly.

  • Page 6, line 175, Figure 6 is Figure 2

This has been corrected accordingly.

  • Page 10 line 289, authors mentioned Hapert’s work. However, such name does not exist in bibliography chapter.

This has been corrected accordingly

  • Page 10, line 291, authors indicated ref 291. However, such reference does not exist.

This has been corrected accordingly.

  • Page 4, equation 1, which is the meaning of  lambda4?.

In the text, λl represents the wavelength of the laser. This has been now indicated accordingly.

  • Please, check the text, figures, constants, etc., in order to correct all errors and then facilitate to review and the understanding of the paper.

This has been done accordingly.

  • Figure 1 presents the Raman spectra of the C-11 sample at temperatures between 1000ºC and 1800ºC. They have performed a spectra deconvolution. Which type of band are selected for deconvolution, Gaussian, Lorentzian, Voight, mixed, etc.?? Please, indicate it in the text.

All bands were deconvoluted using Lorentzian function. This has been mentioned on Page 15 of the manuscript.

  • On the other hand, when figure 1 is examined, baseline is below the experimental data. Then, is the baseline correctly subtracted?

Yes, the baseline was indeed subtracted correctly. The offset was integrated on purpose for a better depiction of the D’ band.

  • Authors performed deconvolution over almost all Raman bands. However, the article entitled “Evolution of the Raman Spectrum with the Chemical Composition of Graphene Oxide”, J. Phys. Chem. C 2017, 121, 20489−20497 DOI: 10.1021/acs.jpcc.7b06236, presents a nice deconvolution that can be taken into account. At least, authors have to include the band located between D and G bands (D’’ band) in order to obtain a better fit of D and G bands.

Thank you for your valuable comment. Indeed, Raman spectra deconvolution of disordered carbon materials was proposed to involve also the D’’ band, which is typically assigned to the amorphous carbon, in addition to those which we considered for the deconvolution of our spectra. However, we think that the trends related to the graphitization of the carbon phase (i.e., decrease/healing of defects and increase of crystallinity) described in the manuscript, are well described by the evolution of the D and G bands, on the one side, as well as of the overtone modes, 2D and D+G, on the other side. We have cited the mentioned paper in our revised manuscript.

  • In page 4, line 135, authors indicated that even the graphitization advances, the intensity of the G band does not. How do they know that the graphitization advances? Please, reference such affirmation. On the other hand, as suggestion, authors can carry out XRD analysis in order to check if the d parameter decreases or increases. Also, TEM analysis can help. Please, check the article entitled “Comparative XRD, Raman, and TEM Study on Graphitization of PBO-Derived Carbon Fibers” J. Phys. Chem. C 2012, 116, 257–268 DOI: /10.1021/jp2084499.

Thank you for your comment. When we discuss about the graphitization evolution in the studied samples, we mainly rely on the appearance and evolution of the overtone modes (e.g. 2D). The 2D band, which in emerges in C-11 (Figure 1) starting with a thermal treatment at 1400 °C has been detected only when the concentration of defects in the carbon phase is rather low. Thus, the term graphitization in our case is focused on healing the defective structure in the segregated sp2-hybridized carbon phase. We have published previously a case study on this aspect, see J. Ceram. Soc. Japan 2016, 124, 1042-1045, in which we have carefully addressed the high temperature evolution of the sp2 carbon phase in SiOC by using UV and visible light Raman spectroscopy. The paper suggested by the Reviewer as well as our own paper have been cited in the revised manuscript.

  • Figure 3 present data from La as a function of the amount of carbon. For example, for SiOC samples containing about 10% of sp2 carbon in vol%, La is about 8.8 nm. Error bars is about ±2 nm. In my opinion, it means that the error in the measurement is high. Does it mean that your sample is not homogeneous and depending of the analysis area you can obtain different results?

Thank you for your valuable comment. Indeed, one should consider and pay attention to possible inhomogeneous distribution of the carbon phase within the matrix – the Raman measurements were done using a grating of 600 and a confocal microscope (magnification 50x, NA = 0.5) with a 100 μm aperture – these parameters provide a resolution of approximately 2 – 4 µm. At this resolution, the occurrence of homogeneities in the samples may be detected only if those are in the same range or larger than the resolution mentioned above. We cannot exclude possible inhomogeneities at micrometer scale in our samples, however, when it comes to assess the macro-samples, we do not expect a strong effect thereof. The rather larger error bars of the measurements may come, in addition to possible inhomogeneities, also from slightly varying conditions of the measurements – for instance, laser beam focus on sample (which e.g. correlates to the sample surface planarity, surface roughness etc.).

  • Page 6, line188, authors said: “…which means that at least one defect is present within the average lateral size of the crystallite…” Please, can you explain such affirmation? What do you mean “with at least one defect”?

The value of LD gives the distance between two defects in the carbon phase. In the mentioned statement, we do compare the LD value for our sample, which is 6 nm, with the lateral cluster size, which is 7.5 nm and conclude that there should be (at least) one defect present (actually, a lateral size of 7.5 nm may accommodate max. 2 defects at LD being 6 nm). As we increase the annealing temperature, LD increases, indicating that the number of defects decreases.

  • Page 8, line 248, I think that t value for SiOC/C sample is 4.3 (not 3.8) as it can be seen in Figure 6.

This has been corrected accordingly.

  • Page 8, lines 250-251 authors compare the t results of SiOC/C with RuO2/silica composites. Which are the similarities between both materials? Do they have the same structure, etc.? Explain it, please.

We compare our samples with RuO2/silica composites, as those also possess a microstructure consisting of a conducting phase (RuO2) dispersed within an insulating matrix (silica). Additionally, the mentioned composites exhibit piezoresistive behavior, which was correlated in the cited paper with the critical exponent t.

  • Figure 5, in page 8, red dots blue rhomb and dark blue squares are from 1600, 1400 and 1100ºC heat treatments. Table 2 shows the SiOC/C selected samples. In such table there are 8 samples. However, in figure 5 you can find 10 red dots. Does it mean that some results related to electrical conductivity for 1600ºC samples are repeated?

There is a total number of 15 prepared samples in the present work. Various samples with a sp²-carbon concentration lower than 20% (C-0, C-1, C-2, C-4, C-6, C-8, C-11) in order to have a precise information about the percolation threshold in this system. Thus, the 10 data points for the samples prepared at 1100 °C are indeed obtained for 10 individual samples with various carbon contents. The total number of prepared samples is shown in Table 1, which in the revised version of the manuscript is found on page 3 (has been moved from the Experimental part, which is at the end of the manuscript, to the Results and Discussion section).

  • Page 9, lines 279-280, authors said: “Interestingly, the activation energy of the C-11 ceramic sample….., i.e. 0.04 eV, is comparable to that of glassy carbon (0.03 eV)”. Why is it interesting? Can you explain? On the other hand, I found articles with different materials such PAN in which the activation energy is similar.

The observation is indeed interesting, as it indicates that the carbon phase, being mainly responsible for the charge carrier transport in the mentioned sample, has same behavior as glassy carbon. So, we may conclude that the segregated carbon phase in our sample has the same conduction behavior as glassy carbon. Obviously, there may be other types of materials probably showing similar activation energies; however, as we do have carbon being segregated in our samples, we compared it with other carbon materials.

  • Page 10, line 288, authors indicated that the conduction occurs within the amorphous silica due to the low carbon concentration present in the SiOC matrix. However, they do not present any reference or if they do not find any, try to explain.

This has been revised accordingly.

  • Page 11, authors mentioned that electrical behavior of SiOC/C samples can be correlated with the concentration of carbon precipitations. They said that results obtained from La is in excellent agreement with values obtained from conductivity measurements. How do you calculate such result? From which plot? Which fitting? It is important to explain in order to facilitate the experimentation for other researchers.

The estimation of the activation energies from the Raman spectroscopic data was performed by using the lateral crystallite size (which has been determined by using eq. (1) from page 6. We did used the eq. Ea = 2.1/La (for carbon-based semiconductors, as reported in ref. 71) for samples with relatively low La (i.e., < 7.5 nm) and the eq. Ea = 7.7/La (used for carbons having high graphitization degree, see ref. 72) for our samples having La values larger than 7.5. This has been shortly described in the manuscript on page 12.

  • It is very tentative to challenge to say that the activation energy in regime III calculated form sp2 carbon is in the same order as the obtained for other methods. From your method Ea= 0.085 eV and other methods Ea=0.035-0.02 eV. The difference is almost 2.5 times higher. In my opinion is a high difference.

We do agree. However, as pointed out in our response above, the estimation is really rough. We do not mention in the revised manuscript anymore that there is a good agreement between the values.

  • In different plots, some data is presented with error bars, however other data do not have error bars. Why? Does it mean that some data are obtained with errors?

In the plots which do not exhibit visible error bars, the size of the error bars is smaller than the size of the data point symbols.

  • The Hall effect is the movement of charge carriers through a conductor towards a magnetic attraction. It means that such effect can be attributed to an electrically conductor material. I think that as the carbon content decreases, the conductivity also decreases. Then, if the material does not present any conductivity or is very low, the Hall effect would be difficult to detect or even it is hard to measure.

Indeed, we fully agree to the Reviewer. We were able to measure charge carrier densities and carrier mobilities only for the samples which exhibited larger sp² carbon contents, i.e. larger than 4%. Below this value no trustworthy data could be obtained – due to this reason, in Figure 9, data for samples with carbon contents >4% are shown.

  • Then, the article is interesting but they have to be improved scientifically (maybe DRX, TEM, …) and make some clarifications.

Due to the tight allocated revision time, we were not able to perform additional measurements on our samples. However, various samples among those studied in the present work (e.g., C-1, C-11, C-17 etc.) were investigated previously with respect to their structural features and the results have been published in various papers (J. Am. Ceram. Soc. 2013, 96, 272; Adv. Func. Mater. 2014, 24, 4097; J. Eur. Ceram. Soc. 2016, 36, 3747; J. Ceram. Soc. Japan 2016, 124, 1042-1045; Adv. Eng. Mater. 2019, 21, 1800596). We thank the reviewer for his comments and suggestions which allowed us to perform the requested clarifications.

Reviewer 3 Report

The paper studies the correlations between the amount and crystallinity of the free carbon phase in silicon oxycarbides and their charge carrier transport behavior. The detailed structural characterization of the segregated carbon phase, electrical conductivity as well as Hall effect measurements are introduced and discussed within the context of determined regimes of charge carrier transport in SiOC/C. The data are well supported with measurements and correlated with literature references. 

Questions:

  1. Where are the segregated carbon can be found in the structure?
  2. How the structure is composed and how is developing druing heat treatments?
  3. Considering the uniaxial pressure of sintering (SPS) texturing of samples may happen, the segregation of phases can be also effected. Can the authors make explanation about this possible structure-property relation?

Author Response

Thank you for the valuable Reviewers comments. We have seriously considered the Reviewers comments and revised our manuscript with respect to their suggestions. All changes performed in the text are highlighted.

The paper studies the correlations between the amount and crystallinity of the free carbon phase in silicon oxycarbides and their charge carrier transport behavior. The detailed structural characterization of the segregated carbon phase, electrical conductivity as well as Hall effect measurements are introduced and discussed within the context of determined regimes of charge carrier transport in SiOC/C. The data are well supported with measurements and correlated with literature references.

Questions:

  • Where are the segregated carbon can be found in the structure?

Typically, segregated carbon has been homogeneously dispersed throughout the investigated samples. Various samples among those investigated in our present work were studied by means of e.g. TEM in order to reveal the presence and dispersion of the segregated carbon phase, such as in the example below:

  • How the structure is composed and how is developing during heat treatments?

The evolution of the phase composition and microstructure of the prepared nanocomposites was studied in previous work published by our group as well as other groups. Some selected published papers within thin context are found in the following: J. Am. Ceram. Soc. 2013, 96, 272; Adv. Func. Mater. 2014, 24, 4097; J. Eur. Ceram. Soc. 2016, 36, 3747; J. Ceram. Soc. Japan 2016, 124, 1042-1045; Adv. Eng. Mater. 2019, 21, 1800596.

  • Considering the uniaxial pressure of sintering (SPS) texturing of samples may happen, the segregation of phases can be also affected. Can the authors make explanation about this possible structure-property relation?

Thank you for your interesting point. We did investigate the possible effect of the SPS on the texturing of the sample or carbon arrangement previously. However, our conclusion was that the SPS conditions (i.e., the uniaxial loading) do not have any influence on this.